# A Tri-Component (Glomerular, Tubular, and Metabolic) Assessment of Renal Function in Acute Heart Failure

**DOI:** 10.3390/jcm13247796

**Published:** 2024-12-20

**Authors:** Gracjan Iwanek, Barbara Ponikowska, Husam Salah, Marat Fudim, Mateusz Guzik, Robert Zymliński, Krzysztof Aleksandrowicz, Beata Ponikowska, Jan Biegus

**Affiliations:** 1Institute of Heart Diseases, Wroclaw Medical University, 50-367 Wrocław, Poland; mateuszguzik23@gmail.com (M.G.); robertzymlinski@gmail.com (R.Z.); krzysztof.aleksandrowicz@umw.edu.pl (K.A.); janbiegus@gmail.com (J.B.); 2Student Scientific Organization, Wroclaw Medical University, 50-367 Wrocław, Poland; ponikowska.b@gmail.com; 3Department of Medicine, Duke University, Durham, NC 27708, USA; husam.salah@duke.edu (H.S.); marat.fudim@duke.edu (M.F.); 4Duke Clinical Research Institute, Durham, NC 27701, USA; 5Department of Physiology and Pathophysiology, Wroclaw Medical University, 50-367 Wroclaw, Poland; beata.ponikowska@umw.edu.pl

**Keywords:** acute heart failure, bicarbonate, natriuresis, outcome, renal function

## Abstract

**Background**: Despite the prevalence of impaired renal function in acute heart failure (AHF) patients, the intricate relationship between glomerular, tubular, and metabolic renal function remains unexplored. We aimed to investigate the co-occurrence of glomerular, tubular, and metabolic renal dysfunction in AHF and their impact on prognosis. **Methods**: eGFR, spot urine sodium, and HCO^3−^ were measured in 243 patients hospitalized for AHF. The population was stratified by the 4-point renal dysfunction score and linked with outcomes. **Results**: Glomerular dysfunction exhibited an elevated risk of death (HR of 2.04; 95% CI [1.24–3.36]; *p* = 0.006), combined risk of death, and HF rehospitalization (HR of 2.03; 95% CI [1.34–3.05]; *p* = 0.005). Similarly, tubular dysfunction correlated with a higher death risk (HR of 1.72; 95% CI [1.04–2.82]; *p* = 0.03) and a higher combined risk (HR of 1.82; 95% CI [1.21–2.74]; *p* = 0.004). While renal metabolic dysfunction was linked to increased death risk (HR of 1.82; 95% CI [1.07–3.11]; *p* = 0.028), it was not associated with composite risk (HR of 1.37; 95% CI [0.88–2.15]; *p* = 0.174). Multivariate analysis revealed a direct association between the renal dysfunction score and death risk (HR of 1.92 per 1 point; 95% CI [1.47–2.52]; *p* < 0.0001) and the combined risk of death and HF rehospitalization (HR of 1.78 per 1 point; 95% CI [1.43–2.22]; *p* < 0.0001). **Conclusions**: Renal dysfunction is common, with varied overlaps. Glomerular, tubular, and metabolic dysfunctions predict adverse outcomes in AHF. The established renal score may aid patient stratification and prognosis.

## 1. Introduction

The complex and bidirectional relationship between the heart and kidneys in heart failure (HF) has been recognized for almost two centuries [1,2,3]. The recognition of this relationship led to the introduction of the term cardiorenal syndrome (CRS), which encompasses several subtypes in which dysfunction of the heart and/or the kidneys results in acute and/or chronic dysfunction of the other organs [2]. Impaired renal function is prevalent among patients hospitalized for acute heart failure (AHF), with >60% prevalence on admission [4], and is independently predictive of all-cause mortality, cardiovascular mortality, and subsequent HF hospitalizations across all HF phenotypes [4,5,6]. The clinical tool to assess renal function in HF has long remained serum creatinine, which is a surrogate biomarker of the glomerular filtration rate (GFR). However, the nephron’s function is far more complex and extends beyond glomerular function alone. Water and sodium handling, which are a reflection of the tubular function in the kidney, is another useful prognosticator in HF [7,8]. Decreased urinary sodium excretion has been associated with a diuretic response and poor outcomes [9,10,11], making spot urinary sodium measurement a useful tool to monitor the effectiveness of decongestive therapies as well as an important prognostic tool [8,10,11,12]. Therefore, spot urine sodium levels in patients undergoing diuresis can be considered a surrogate marker of tubular function. A third renal component may prove useful to obtain a more comprehensive, tri-component image of renal function, i.e., assessment of acid–base balance, with HCO_3_^−^ levels reflecting the renal control of this metabolic balance. Herein, we assess the association between the kidney’s glomerular, tubular, and metabolic function (using serum creatinine, spot urine sodium, and HCO_3_^−^ as surrogate markers for these functions, respectively) and the outcomes of patients hospitalized for AHF.

## 2. Materials and Methods

This is a single-center, observational study based on a prospective registry that was run from January 2016 to September 2017 in the 4th Military Hospital in Wroclaw. Study participants were hospitalized adults (>18 years old) who had AHF as the main reason for their hospitalization and received intravenous furosemide upon admission. Patients with cardiogenic shock, acute coronary syndrome, severe liver disease, and/or end-stage renal disease requiring renal replacement therapy were excluded from the study. In this study, AHF was defined based on the guidelines of the European Society of Cardiology. All patients provided written consent, and the study was approved by the local ethics committee (approval No. KB-387/2015) and conducted in compliance with the Declaration of Helsinki and Good Clinical Practice.

### 2.1. Study Procedures

After being admitted to the hospital, the participants underwent a clinical examination, during which detailed information about their demographics (including their history of HF), comorbidities, previous treatment(s), and physical examination findings were recorded. At each predefined time-point (admission, day^−1^), the participants’ blood and urine samples were taken for analyses. Some samples were collected, centrifuged, frozen (at −70 °C), and stored for further assessments. The study measured various laboratory parameters using standard methods in the local laboratory. Plasma concentrations of N-terminal pro–B-type natriuretic peptide (NTproBNP) and cardiac troponin were measured using immunoenzymatic methods. NTproBNP levels were determined with an immunoenzymatic assay (Siemens, Marburg, Germany), while cardiac troponin concentrations were analyzed using the single Dimension RxL Max system (Siemens). The renin and aldosterone system activation was measured using a chemiluminescent immunoassay-CLIS, LIAISON from the initially frozen samples.

### 2.2. Renal Function Assessment

Glomerular function was assessed by eGFR (calculated using the MDRD (Modification of Diet in Renal Disease formula)) on admission, with a cutoff value of <60 mL/min/1.73 m^2^ [8,13] defining glomerular dysfunction. The tubular function was assessed by a spot urine sodium examination on day^−1^ of hospitalization, with a cutoff value of ≤60 mmol/L defining tubular dysfunction. The metabolic function of the kidneys was assessed by the circulating HCO_3_^−^ levels using peripheral blood gas on admission, with a cutoff value of <21 mmol/L [14] (the lower limit of normal) defining metabolic dysfunction of the kidneys.

### 2.3. Categorization

A scoring system was created to assess renal dysfunction, consisting of glomerular, tubular, and metabolic dysfunction. Each type of dysfunction was assigned a point, and the total score was determined by summing these values. The overall score for renal dysfunction ranged from 0 (indicating no dysfunction) to 3 (indicating abnormalities in all of the three analyzed components of renal function).

### 2.4. Study Outcomes

The clinical endpoints of the study were:In-hospital mortality;One-year all-cause mortality;Composite endpoint of one-year all-cause mortality and rehospitalization for the HF.

### 2.5. Clinical Follow-Up

Participants who were discharged from the hospital were closely monitored, as per the protocols of the HF clinic, for a minimum of one year. Multiple sources were utilized to gather information regarding participants’ survival status and readmission to the hospital, including patient feedback, interviews with their family members over the phone, relevant clinic databases, and/or the National Registery of Citizens. No participants were lost to follow-up.

### 2.6. Statistical Analysis

Continuous variables with a normal distribution were presented as a mean ± standard deviation, while variables with a skewed distribution were described using medians with upper and lower quartiles. To demonstrate differences between study groups we employed ANOVA for variables with a normal distribution, the Mann–Whitney U-test for skewed variables, and ꭓ^2^ test for categorical variables. Cox proportional hazard models were used to calculate the hazard ratio (HR) with corresponding 95% confidence intervals (95% CI) for all-cause mortality. The multivariable analysis was adjusted for confounding variables, including systolic blood pressure on admission, left ventricle ejection fraction, hemoglobin, troponin I, and serum Na^+^ concentration on admission. The continuous variables included in the multivariable Cox model were assessed for their functional form by testing the proportional hazard assumption and visually examining the Schoenfeld residuals. Any missing data in the multivariable model were imputed using the mean value. For categorical variables, missing values were substituted with the most frequently occurring category. Kaplan–Meier survival curves were used to visualize the survival analysis. Multivariate regression models were developed to illustrate the clinical and laboratory determinants of renal dysfunction on admission. A *p*-value less than 0.05 was considered statistically significant. The statistical analysis was performed using STATISTICA 13 (StatSoft).

## 3. Results

### 3.1. Baseline Characteristics

Baseline characteristics are summarized in Table 1. The study population included 243 patients (predominately male [73.3%]) with a mean age of 70 ± 12.7 years and mean LVEF of 37 ± 14%. An ischemic etiology of HF was present in 123 (50.6%) of the patients, and 102 (42%) had de novo AHF. The study population had a mean creatinine value of 1.4 ± 0.5 mg/dL, mean spot urine sodium value of 76 ± 38 mmol/L, and mean HCO_3_^−^ value of 23.3 ± 3.7 mmol/L. On admission, the median NTproBNP level was 5618.0 [3431.0–11749.5] pg/mL and the mean lactate level was 2.2 ± 1.1 mmol/L.

### 3.2. Renal Dysfunction Score

Participants were stratified into four groups based on their renal dysfunction score on admission. The stratification was as follows: 86 patients (35.4%) had a renal dysfunction score of 0, 106 patients (43.6%) had a renal dysfunction score of 1, 41 patients (16.9%) had a renal dysfunction score of 2, and 10 patients (4.1%) had a renal dysfunction score of 3. At baseline, there were no significant differences among the groups in terms of sex (*p* = 0.273), age (*p* = 0.154), LVEF (*p* = 0.349), or etiology of HF (*p* = 0.184) (Table 2).

On admission, the group with a renal dysfunction score of 0 had the lowest levels of NTproBNP (a median of 5047 pg/mL [3100–8668]) and lactate (a mean of 1.9 ± 0.5 mmol/L), while the group with a renal dysfunction score of 3 had the highest levels of NTproBNP (a median of 11,608 pg/mL [5080–18,580]) and lactate (a mean of 3.7 ± 2.8 mmol/L). The group with a renal dysfunction score of 2 and 3 had a significantly higher length of stay (11.6 ± 8.7 and 9.5 ± 3.8 days, respectively) compared with the other groups (7.1 ± 3.5 days in the renal dysfunction score of 0; 7.5 ± 5.2 days in the renal dysfunction score of 1; *p* < 0.001). Patients with bi- and tri-component renal dysfunction were more likely to reach the prespecified clinical endpoints (Table 3).

### 3.3. The Prevalence and Co-Existence of Different Types of Renal Dysfunctions

In bi-component renal dysfunction, glomerular and tubular dysfunction were present in 14 patients (34%), glomerular and metabolic dysfunction in 14 patients (34%), and tubular and metabolic dysfunction in 13 patients (32%). Glomerular dysfunction alone was evident in 30 patients (28%), while tubular dysfunction was identified in 56 patients (53%), and metabolic dysfunction was discerned in 20 patients (19%). (Appendix A).

### 3.4. The Associates of Renal Dysfunction in AHF

In the multivariable model, the age, aldosterone level, and NTproBNP level were independently associated with glomerular dysfunction on admission. Tubular dysfunction was associated with aldosterone alone, while metabolic dysfunction was associated with elevated NTproBNP alone, all *p* < 0.05 (Table 4). The sensitivity analysis for serum creatinine concentrations on admission is presented in Appendix A.

### 3.5. Impact of Renal Dysfunction on Prognosis

#### 3.5.1. Glomerular Dysfunction

In the multivariate model, eGFR was inversely associated with the risk of death (HR of 0.986 per 1 mL/min/1.73 m^2^; 95% CI [0.977–0.996]; *p* = 0.006) and the composite risk of death and HF rehospitalization (HR of 0.984 per 1 mL/min/1.73 m^2^; 95% CI [0.976–0.992]; *p* < 0.0001). Glomerular dysfunction, defined as eGFR < 60 mL/min/1.73 m^2^ on admission, was associated with an increased risk of death (HR of 2.04; 95% CI [1.24–3.36]; *p* = 0.005) and an increased composite risk of death and HF rehospitalization (HR of 2.03; 95% CI [1.34–3.05]; *p* < 0.001) (Table 5).

#### 3.5.2. Tubular Dysfunction

In the multivariate model, spot urine sodium was inversely associated with the risk of death (HR of 0.99 per 1 mmol/L; 95% CI [0.98–1.00]; *p* = 0.004) and the composite risk of death and HF rehospitalization (HR of 0.99 per 1 mmol/L; 95% CI [0.99–1.00]; *p* = 0.002). Tubular dysfunction, defined as spot urine sodium ≤ 60 mmol/L at day^−1^, was associated with an increased risk of death (HR of 1.72; 95% CI [1.04–2.82]; *p* = 0.03) and an increased composite risk of death and HF rehospitalization (HR of 1.82; 95% CI [1.21–2.74]; *p* = 0.004) (Table 4).

#### 3.5.3. Renal Metabolic Dysfunction

In the multivariate model, serum bicarbonate was inversely associated with the risk of death (HR of 0.93 per 1 mmol/L; 95% CI [0.86–0.99]; *p* = 0.03) and the composite risk of death and HF rehospitalization (HR of 0.94 per 1 mmol/L; 95% CI [0.89–0.99]; *p* = 0.031). Renal metabolic dysfunction, defined as serum bicarbonate level <21 mmol/L on admission, was associated with an increased risk of death (HR of 1.82; 95% CI [1.07–3.11]; *p* = 0.028) but not associated with the composite risk of death and HF rehospitalization (HR of 1.37; 95% CI [0.88–2.15]; *p* = 0.174) (Table 4).

### 3.6. Prognostic Value of the Renal Dysfunction Score

In the multivariate model, the renal dysfunction score was directly associated with the risk of death (HR of 1.87 per 1 point; 95% CI [1.38–2.54]; *p* = 0.0005) and the composite risk of death and HF rehospitalization (HR of 1.71 per 1 point; 95% CI [1.35–2.19]; *p* = 0.0001) (Table 5, Figure 1 and Figure 2). Kaplan–Meier curves for direct serum creatinine assessment, incorporated in the renal dysfunction score, are presented in Appendix A. Associations between covariates used for adjustment in the multivariable model are presented in the Appendix A.

## 4. Discussion

The current analysis shows that, in patients with AHF, glomerular dysfunction, tubular dysfunction, and renal metabolic dysfunction are independently associated with worse prognoses in terms of death and the composite risk of death and rehospitalization for HF. The study also shows that a simple scoring tool that assigns one point to each type of renal dysfunction (assessed using commonly measured surrogates) can predict long-term outcomes in and the length of stay for patients with AHF. Furthermore, the scoring tool demonstrates notable promise in the realm of tailoring clinical management [15] and optimizing diuretic therapy [16]. The mechanism of renal dysfunction in AHF is complex, with several involved pathways. While, traditionally, the predominant pathway of renal dysfunction in AHF has been attributed to impaired cardiac output and the relative underfilling of arterial perfusion—with subsequent activation of the renin–angiotensin–aldosterone system and sympathetic nervous system as well as an increase in arginine vasopressin secretion [17,18]—emerging evidence suggests that central and renal venous and lymphatic congestion may be more important contributors to the pathophysiology of renal dysfunction in AHF [19,20]. These factors, individually or cumulatively, can result in renal interstitial edema with a subsequent increase in intrarenal pressure (due to the rigidness of the renal capsule), reduction in the urinary flow and urinary sodium concentration, and a renal parenchymal hypoxic state [18,21].

Consistent with previous analyses [4,5,6], the current study shows that glomerular dysfunction on admission (as assessed by elevated serum creatinine level/eGFR) is associated with worse outcomes in patients hospitalized for AHF. Our study expands on this knowledge and shows that other types of renal dysfunction (i.e., tubular and metabolic dysfunction) are also individually associated with worse outcomes and should be considered when evaluating patients with AHF. Of note, the assessment of metabolic function is not a standard of care recommended in AHF patients, although it provides important clinical information. The current study also shows that different types of renal dysfunction, if present concurrently, may have additive effects and be associated with worse outcomes. It is important to understand that the current study evaluates glomerular dysfunction on admission as worsening glomerular function (i.e., rising creatinine) during decongestive therapy (i.e., later during admission), which is not necessarily associated with worse outcomes in those with a good diuretic response [22,23].

In addition to the prognostic value of glomerular dysfunction in HF, the prognostic value of tubular dysfunction has been demonstrated in prior studies. In an analysis of the GISSI-HF trial, tubular injury in patients with stable chronic HF—which was assessed using urinary markers of tubular damage, such as N-acetyl-beta-D-glucosaminidase (NAG), kidney injury molecule 1 (KIM-1), and neutrophil gelatinase-associated lipocalin (NGAL)—was associated with all-cause mortality and HF hospitalization. [24] Also, higher tubular marker concentrations were seen in those with more severe disease (e.g., advanced NYHA functional class, more diuretic use) [24]. However, these markers are not readily available in clinical practice, which may limit their application in routine clinical practice. In addition, a direct association between tubular dysfunction/injury and outcomes in patients with AHF has not been previously investigated. In the current study, we use spot urine sodium at day^−1^ of hospitalization for AHF as a marker of tubular function. Low urine sodium excretion in patients with AHF has long been recognized as a marker of poor diuretic response [25], and decreased tubular delivery of diuretics was suggested as a main mechanism driving this association. However, an accumulating body of evidence suggests that the association between low urine sodium excretion and poor diuretic response is related to a blunted tubular response and tubular dysfunction rather than decreased tubular delivery [25,26]. This knowledge positions urine sodium as a surrogate marker of tubular function in patients undergoing diuretic therapy.

Prior analyses investigating the association between serum HCO_3_^−^ level on admission and outcomes in patients hospitalized for AHF are sparse. In one cohort of patients hospitalized for AHF and stratified into terciles based on the mean of the two HCO_3−_ values (mean HCO_3_^−^ ≤ 26, 26 < mean HCO_3_^−^ ≤ 28, and mean HCO_3_^−^ > 28), there was no significant association between HCO_3_^−^ levels on admission and the primary outcome (i.e., time to death) or secondary outcomes (i.e., time to first HF rehospitalization and the composite of death or HF rehospitalization) [14]. In contrast, our study shows that a lower HCO_3_^−^ level on admission is associated with worse outcomes in AHF. This discrepancy may stem from different populations of AHF, as our population had a mean HCO_3_^−^ of 23, which may suggest greater renal metabolic dysfunction in our population compared to the prior study.

### Limitations

Several limitations should be acknowledged. First, this study was an observational, single-center study, which may limit the generalizability of its results. Second, we used surrogate markers to assess glomerular dysfunction, tubular dysfunction, and renal metabolic dysfunction. While serum creatinine is a commonly used and validated marker of glomerular function, the utility, sensitivity, and specificity of spot sodium urine and serum bicarbonate level as markers of tubular and renal metabolic functions, respectively, are less validated. Third, the tubular and metabolic functions of the kidneys extend beyond handling sodium and bicarbonate (e.g., metabolism and reabsorption of amino acids, glucose, potassium, phosphate, and albumin) [27,28], which were not fully evaluated in the current study.

The metabolic function of a kidney are complex and multidimensional and cannot be adequately reflected by any single biomarker, including serum bicarbonate. Similarly, the glomerular and tubular function of the kidney cannot be adequately captured by the biomarkers utilized in the scoring system.

## 5. Conclusions

Renal dysfunction is prevalent, and the overlap between different types of renal dysfunction is frequent. Glomerular, tubular, and renal metabolic dysfunction are independently associated with worse prognoses, including death and HF rehospitalizations, in patients with AHF. A simple scoring tool for renal dysfunction can be used to better stratify patients admitted for AHF and improve prognostication.

## Figures and Tables

**Figure 1 jcm-13-07796-f001:**
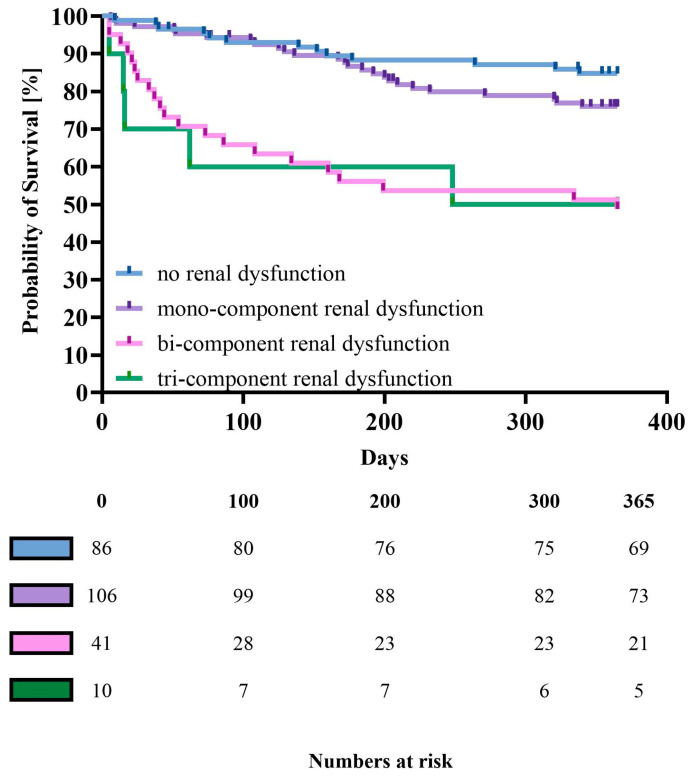
Kaplan–Meier curves for one-year mortality by the renal dysfunction score. Log-rank, *p* < 0.0001.

**Figure 2 jcm-13-07796-f002:**
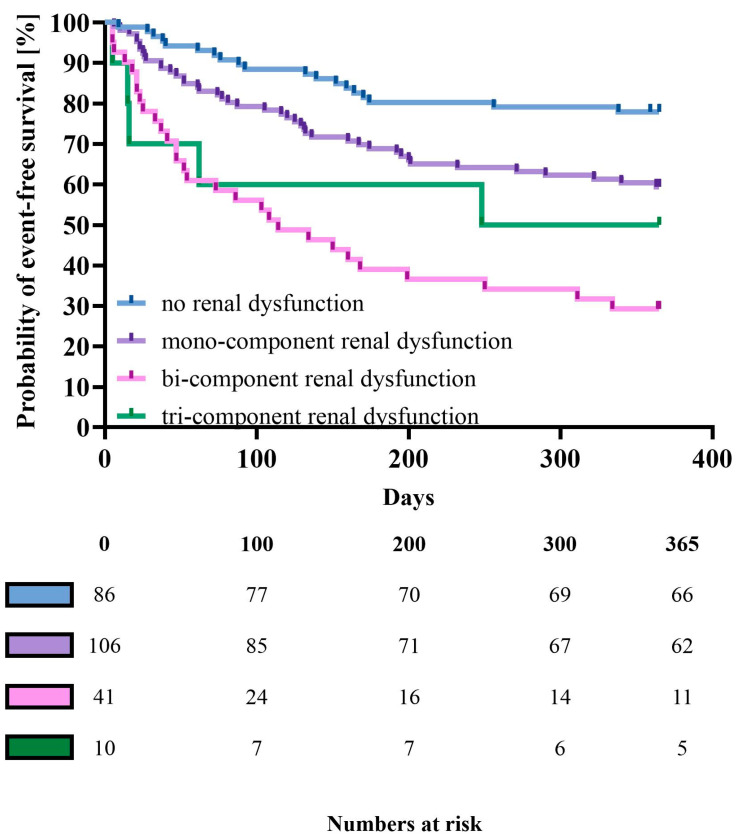
Kaplan–Meier curves for death or heart failure rehospitalization (whichever occurred first) by the renal dysfunction score. Log-rank, *p* < 0.0001.

**Table 1 jcm-13-07796-t001:** Baseline characteristics of the population.

Parameter	Population
Number of patients	243
Sex, female	65 (26.7%)
Age, years	70.0 ± 12.7
NYHA classification	
III	60 (25%)
IV	183 (75%)
Heart rate, beat/minute	90.8 ± 24.3
Systolic blood pressure on admission, mmHg	134.0 ± 31.2
Diastolic blood pressure on admission, mmHg	79.0 ± 16.2
Left ventricle ejection fraction, %	37.1 ± 13.8
Acute heart failure, de novo	102 (42%)
Acute heart failure etiology	
Suboptimal therapy	58 (24%)
Infection	42 (17%)
Tachyarrhythmia	36 (15%)
Chronic heart failure progression	34 (14%)
Noncompliance	27 (11%)
Hypertensive crisis	24 (10%)
Myocarditis	10 (4%)
Unknown	29 (12%)
Comorbidities	
Coronary artery disease	137 (56%)
Hypertension	194 (80%)
Chronic kidney disease	115 (47%)
Valvular disorders	143 (59%)
Dyslipidemia	131 (54%)
Arrhythmias	137 (56%)
Implantable devices	
Pacemaker	18 (7%)
Cardioverter-defibrillator	35 (14%)
Cardiac resynchronization therapy with a defibrillator	14 (6%)
Blood count	
Hemoglobin, g/dL	13.3 ± 2.0
WBC, G/L	9.3 ± 4.5
PLT, G/L	210.1 ± 88.5
Iron metabolism	
Iron, μg/dL	56 ± 29.9
TIBC, µg/dL	347 ± 69
UIBC, µg/dL	291 ± 70
Transferrin saturation, %	16.4 ± 9
Soluble transferrin receptor, mg/L	2.0 ± 0.8
Ferritin, ng/mL	175 ± 147
Renal function	
Creatinine on admission, mg/dL	1.4 ± 0.5
eGFR on admission, mL/min/1.73 m^2^	78 ± 28
Blood urea nitrogen, mg/dL	59.5 ± 30.9
Spot urine sodium, mmol/L	76.2 ± 37.6
HCO_3_^−^, mmol/L	23.3 ± 3.7
Renin, µIU/mL	10.6 (3.1–104.7)
Aldosterone, ng/dL	10.5 (6.6–17.6)
AST, IU/L	28.0 (21.5–41.0)
ALT, IU/L	30.5 (21.0–56.0)
Bilirubin, mg/dL	1.02 (0.72–1.71)
Na, mmol/L	139.0 ± 4.3
CRP, mg/L	7.6 (4.1–18.9)
NTproBNP, pg/mL	5618.0 (3431.0–11,749.5)
Troponin I, ng/mL	0.057 (0.025–0.160)
Creatinine at day^−1^, mg/dL	1.3 ± 0.5
Creatinine at discharge, mg/dL	1.2 ± 0.5
Lactate on admission, mmol/L	2.2 ± 1.1
Lactate at day^−1^, mmol/L	2.1 ± 1.2
Length of hospitalization, days	8.1 ± 5.6
Baseline medication	
ACEI/ARB	106 (44%)
Beta-blockers	138 (57%)
Aldosterone antagonist	65 (27%)
Loop diuretics	119 (49%)

ACEI—angiotensin-converting enzyme inhibitors; ARB—angiotensin 2 receptor blockers; ALT—alanine aminotransferase; AST—asparagine aminotransferase; CRP—C-reactive protein; eGFR—estimated glomerular filtration rate; PLT—platelet count, TIBC—total iron binding capacity; UIBC—unsaturated iron binding capacity; WBC—white blood count.

**Table 2 jcm-13-07796-t002:** Comparison of baseline characteristics between patients with different profiles of renal dysfunction.

Profile of Renal Dysfunction (Number of Renal Dysfunctions)	No Renal Dysfunction	Mono-Component Renal Dysfunction	Bi-Component Renal Dysfunction	Tri-Component Renal Dysfunction	*p*-Value
Parameter					
Number of patients	86	106	41	10	
Sex, female	19 (11%)	24 (23%)	17 (41%)	5 (50%)	0.273
Age, years	69.2 ± 12.1	69.14 ± 13.1	72.7 ± 12.9	76.4 ± 10.4	0.154
Heart rate, beat/minute	88 ± 24	93 ± 24	88 ± 24	97 ± 21	0.393
Systolic blood pressure on admission, mmHg	136.9 ±29.6	134.5 ± 31.7	129.1 ± 32.0	122.1 ± 34.8	0.359
Diastolic blood pressure on admission, mmHg	81.1 ± 14.9	80.1 ± 16.9	73.7 ± 15.7	70.5 ± 16.63	0.028
Left ventricle ejection fraction, %	35 ± 12	36 ± 13	40 ± 15	38 ± 15	0.349
Acute heart failure, de novo	34 (40%)	51 (48%)	12 (29%)	5 (50%)	0.184
Ischemic etiology of heart failure	42 (49%)	51 (48%)	23 (56%)	7 (70%)	0.66
Blood count					
Hemoglobin, g/dL	13.6 ± 1.8	13.5 ± 1.8	12.6 ± 2.1	12 ± 2.98	0.008
WBC, G/L	8.8 ± 3.2	9 ± 3.5	10 ± 6.3	14.3 ± 9.1	0.001
PLT, G/L	215 ± 89	197.8 ± 76.8	220 ± 89	264 ± 158	0.077
Iron metabolism					
Iron, μg/dL	58.9 ± 26.4	57.1 ± 27.3	52 ± 41.5	28.1 ± 14.8	0.041
TIBC, µg/dL	353 ± 77	350 ± 61	330 ± 63.4	300 ± 62	0.078
UIBC, µg/dL	294 ± 77	293 ± 61	278 ± 73	271 ± 64	0.518
Transferrin saturation, %	17 ± 7.4	16.5 ± 8.7	16.2 ± 12.8	9.7 ± 5.2	0.19
Soluble transferrin receptor, mg/L	1.8 ± 0.6	2 ± 0.8	2.4 ± 0.9	2.4 ± 1.2	0.007
Ferritin, ng/mL	176 ± 148	190 ± 161	138 ± 99	144 ± 123	0.331
Renal function					
Creatinine on admission, mg/dL	1.12 ± 0.23	1.33 ± 0.45	1.75 ± 0.65	2.1 ± 0.79	<0.001
eGFR on admission, mL/min/1.73 m^2^	90 ± 19	80 ± 29	55 ± 19	42 ± 13	<0.001
Blood urea nitrogen, mg/dL	48 ± 17	56 ± 24	80 ± 42	106.7 ± 50.3	<0.001
Spot urine sodium, mmol/L	102.4 ± 30.7	82.8 ± 34.5	63.9 ± 33.8	63.9 ± 33.8	<0.001
HCO_3_^−^, mmol/L	24.9 ± 2.9	23.6 ± 3.1	20.5 ± 4.2	18.5 ± 3.1	0.044
Renin at day^−1^, µIU/mL	16.2 (3.5–74.7)	18.3 (6.3–190)	79.7 (21.9–218.6)	20.5 (6.8–71.2)	0.045
Aldosterone at day^−1^, ng/dL	8.2 (5.8–13.5)	11.1 (6.8–18.2)	15.4 (8.6–35.9)	10.9 (8.1–27.5)	<0.028
AST, IU/L	28 (20–40)	29 (23–41)	27 (22–40)	56 (34–100)	0.828
ALT, IU/L	30 (21–56)	33 (22–41)	27 (20–52)	77 (25–126)	0.401
Bilirubin, mg/dL	0.96 (0.61–1.52)	1.06 (0.75–1.69)	1.21 (0.77–2.16)	0.43 (0.35–1.74)	0.309
Na, mmol/L	140 ± 3.7	139 ± 4.3	136.8 ± 4.9	139.2 ± 4.26	0.002
CRP, mg/L	5.2 (2.8–9.8)	7.3 (4.2–14.1)	12.1 (5.2–29.7)	35.5 (11.6–49.0)	0.045
NTproBNP, pg/mL	5047 (3100–8668)	5297 (3368–10,036)	7611 (5047—17,354)	11,608 (5080–18,580)	0.007
Troponin I, ng/mL	0.04 (0.02–0.10)	0.05 (0.02–0.20)	0.08 (0.04–0.14)	0.26 (0.04–1.1)	0.254
Creatinine at day^−1^, mg/dL	1.12 ± 0.27	1.27 ± 0.45	1.67 ± 0.62	1.8 ± 0.7	<0.001
Creatinine at discharge, mg/dL	1.08 ± 0.25	1.26 ± 0.55	1.46 ± 0.49	1.3 ± 0.7	0.001
Lactate on admission, mmol/L	1.9 ± 0.6	2.2 ± 1	2.6 ± 1.4	3.7 ± 2.8	<0.0001
Lactate at day^−1^, mmol/L	1.9 ± 0.5	2.0 ± 0.7	2.4 ± 1.4	3.4 ± 3.2	<0.005
Length of hospitalization, days	7.1 ± 3.5	7.5 ± 5.2	11.6 ± 8.7	9.5 ± 3.8	<0.001
Baseline medication					
ACEI/ARB	34 (40%)	51 (48%)	14 (37%)	6 (60%)	0.338
Beta-blockers	40 (47%)	66 (62%)	27 (66%)	5 (50%)	0.087
Aldosterone antagonist	19 (22%)	29 (27%)	12 (29%)	5 (50%)	0.277
Loop diuretics	34 (40%)	51 (48%)	41 (66%)	7 (70%)	0.022

ACEI—angiotensin-converting enzyme inhibitors; ARB—angiotensin 2 receptor blockers; ALT—alanine aminotransferase; AST—asparagine aminotransferase; CRP—C-reactive protein; eGFR—estimated glomerular filtration rate; NTproBNP—N-terminal pro-B-type natriuretic peptide; PLT—platelet count, TIBC—total iron binding capacity; UIBC—unsaturated iron binding capacity; WBC—white blood count.

**Table 3 jcm-13-07796-t003:** Prespecified endpoints by the number of renal dysfunctions.

Prespecified Endpoints	Profile of Renal Dysfunction (Number of Renal Dysfunctions)				
	No Renal Dysfunction (*n* = 86)	Mono-Component Renal Dysfunction (*n* = 106)	Bi-Component Renal Dysfunction (*n* = 41)	Tri-Component Renal Dysfunction (*n* = 10)	*p*-Value
In-hospital mortality	1 (1%)	2 (2%)	3 (7%)	3 (30%)	<0.0001
1-year mortality	17 (20%)	33 (31%)	20 (49%)	5 (50%)	0.005
1-year mortality or heart failure rehospitalization (whichever occurred first)	20 (23%)	44 (42%)	30 (73%)	6 (60%)	<0.0001

**Table 4 jcm-13-07796-t004:** Clinical and laboratory determinants of renal dysfunction.

	Multivariate Regression Model; b-Coefficient	*p*-Value
eGFR on Admission < 60 mL/min/1.73 m^2^		
R-value of the model = 0, *p* < 0.0		
Age, years	0.084	0.002
Gender, male	−0.185	0.737
Serum Na^+^, mmol/L	0.050	0.367
Systolic blood pressure, mmHg	0.014	0.167
Left ventricle ejection fraction, %	0.041	0.057
NTproBNP, pg/mL	0.0001	0.002
Bilirubin, mg/dL	0.288	0.208
Lactate, mmol/L	0.102	0.728
Aldosterone at day^−1^, ng/dL	0.037	0.035
UNa^+^ < 60 mmol/L		
R-value of the model = 0, *p* < 0.0		
Age, years	−0.063	0.53
Gender, male	−0.099	0.26
Serum Na^+^, mmol/L	−0.004	0.96
Systolic blood pressure, mmHg	−0.0004	1.0
Left ventricle ejection fraction, %	−0.017	0.86
NTproBNP, pg/mL	−0.091	0.31
Bilirubin, mg/dL	−0.024	0.79
Lactate, mmol/L	0.038	0.69
Aldosterone at day^−1^, ng/dL	0.266	0.008
HCO_3_^−^ < 21 mmol/L		
R-value of the model = 0, *p* < 0.0		
Age, years	0.049	0.61
Gender, male	−0.028	0.74
Serum Na^+^, mmol/L	−0.106	0.26
Systolic blood pressure, mmHg	−0.035	0.7
Left ventricle ejection fraction, %	0.119	0.22
NTproBNP, pg/mL	0.246	0.006
Bilirubin, mg/dL	0.091	0.31
Lactate, mmol/L	0.143	0.12
Aldosterone at day^−1^, ng/dL	0.014	0.88

NTproBNP—N-terminal pro–B-type natriuretic peptide; UNa^+^—urine sodium concentration.

**Table 5 jcm-13-07796-t005:** Impact of renal dysfunction on prognosis. Hazard ratios for 1-year mortality and 1-year mortality or heart failure rehospitalization (whichever occurred first).

	HR (95% CI)	*p*	HR (95% CI)	*p*	ꭓ^2^/*p* for the Model
	Univariate Model		Multivariate Model *		
1-year mortality risk					
eGFR, ml/min/1.73 m^2^ per 1 mL/min/1.73 m^2^	0.986 (0.977–0.996)	0.006	0.987 (0.977–0.998)	0.02	31.2/0.0003
Urine Na, mmol/L, per 1 mol/L	0.99 (0.98–0.99)	0.0005	0.988 (0.980–0.995)	0.001	36.3/0.0004
HCO_3_^−^ akt, mmol/L, per 1 mol/L	0.91 (0.84–0.98)	0.009	0.95 (0.89–0.98)	0.132	27.8/0.001
eGFR < 60 mL/min/1.73 m^2^	2.04 (1.24–3.36)	0.005	1.98 (1.13–3.46)	0.017	31.1/0.0003
Urine Na^+^ at day^−1^ ≤ 60 mmol/L	1.91 (1.17–3.13)	0.009	2.01 (1.20–3.36)	0.008	20.1/0.0002
HCO_3_^−^ < 21 mmol/L	2.16 (1.3–3.61)	0.003	1.68 (0.97–2.91)	0.065	28.7/0.0007
Number of dysfunctions, per 1 dysfunction	1.92 (1.47–2.52)	<0.0001	1.93 (1.40–2.65)	0.00005	41.5/<0.00001
	Univariate model		Multivariate model *		
1-year mortality or heart failure rehospitalization (whichever occurred first)					
eGFR, mL/min/1.73 m^2^ per 1 mL/min/1.73 m^2^	0.984 (0.976–0.992)	<0.0001	0.981 (0.973–0.990)	<0.0001	43.6/<0.00001
Urine Na, mmol/L, per 1 mol/L	0.99 (0.98–0.99)	<0.0001	0.990 (0.984–0.996)	0.001	36.7/0.0003
HCO_3_^−^ akt, mmol/L, per 1 mol/L	0.93 (0.87–0.98)	0.011	0.95 (0.89–1.00)	0.031	29.1/0.0006
eGFR < 60 mL/min/1.73 m^2^	2.03 (1.34–3.05)	0.0007	2.35 (1.48–3.74)	0.0003	38.1/0.00002
Urine Na^+^ at day^−1^ ≤ 60 mmol/L	2.09 (1.40–3.11)	0.0003	1.92 (1.26–2.91)	0.002	37.1/0.00005
HCO_3_^−^ < 21 mmol/L	1.61 (1.04–2.50)	0.031	1.29 (0.81–2.06)	0.174	29.1/0.001
Number of dysfunctions, per 1 dysfunction	1.78 (1.43–2.22)	<0.000001	1.77 (1.38–2.28)	<0.0001	44.9/<0.00001

* Adjusted for age, sex, systolic blood pressure on admission, left ventricle ejection fraction, hemoglobin, NTproBNP, troponin I, and serum Na^+^ concentration. eGFR—estimated glomerular filtration rate; HR—hazard ratio.

## Data Availability

The data presented in this study are available upon reasonable request.

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
