# Peer review of "A Tri-Component (Glomerular, Tubular, and Metabolic) Assessment of Renal Function in Acute Heart Failure"

_jcm, 2024, doi:10.3390/jcm13247796_

Round 1

Reviewer 1 Report

Comments and Suggestions for Authors

Starting from the ideea that the relationship between the heart and kidneys in heart  failure is a very complex one, its recognition led to the introduction of the term cardiorenal syndrome (CRS), which encompasses several subtypes, in which dysfunction of the heart and/or the kidneys results in acute and/or chronic dysfunction of the other organs.

Impaired renal function is prevalent especially among patients hospitalized for acute heart failure (AHF), with > 60% prevalence on admission  and is independently predictive of all-cause mortality, cardiovascular mortality, and subsequent HF hospitalizations across all HF phenotypes.

The aim of the study was to assess the association between the kidneys' glomerular, tubular, and metabolic functions and outcomes in patients hospitalized for AHF.

The current analysis shows that in patients with AHF, glomerular dysfunction, tubular dysfunction, and renal metabolic dysfunction are independently associated with worse prognosis in terms of death and the composite risk of death and rehospitalization for HF.

 The study also shows that a simple scoring tool that assigns one point to each type of renal dysfunction can predict longterm outcomes and length of hospitalization in patients with AHF.

The design of the study is well done, clear.

Pertinent conclusions, with practical applications. A simple scoring tool for renal dysfunction can be used to better stratify patients admitted for AHF and improve prognostication.

Author Response

Reviewer 1

Starting from the ideea that the relationship between the heart and kidneys in heart failure is a very complex one, its recognition led to the introduction of the term cardiorenal syndrome (CRS), which encompasses several subtypes, in which dysfunction of the heart and/or the kidneys results in acute and/or chronic dysfunction of the other organs.

Impaired renal function is prevalent especially among patients hospitalized for acute heart failure (AHF), with > 60% prevalence on admission and is independently predictive of all-cause mortality, cardiovascular mortality, and subsequent HF hospitalizations across all HF phenotypes.

The aim of the study was to assess the association between the kidneys' glomerular, tubular, and metabolic functions and outcomes in patients hospitalized for AHF. The current analysis shows that in patients with AHF, glomerular dysfunction, tubular dysfunction, and renal metabolic dysfunction are independently associated with worse prognosis in terms of death and the composite risk of death and rehospitalization for HF.

The study also shows that a simple scoring tool that assigns one point to each type of renal dysfunction can predict longterm outcomes and length of hospitalization in patients with AHF.

The design of the study is well done, clear.

Pertinent conclusions, with practical applications. A simple scoring tool for renal dysfunction can be used to better stratify patients admitted for AHF and improve prognostication.

Dear Sir or Madame,

Thank you very much for your positive feedback and for taking the time to review our work. We are delighted that you found the study well-designed and informative.

Reviewer 2 Report

Comments and Suggestions for Authors

The authors of the manuscript present results form an original study, including 243 patients with acute heart failure (ACF), in which they measured eGFR, spot urine sodium, and HCO3. The manusript is topical and important from clinical point of view as impaired renal function is common in patients with AHF. Any clinical studies focused on improved evaluation of the renal function in these patients and more precised risk stratification based on it will be appreciated by the clinicians. 

I have the following recommendations to the authors:

1. I suggest slight modification of the title - "three-component" seems to sound better compared to "three-dimensional" in the context of the topic;

2. The aim of the study is not clear in the abstract (actually, it is missing). 

3. The concomitant risk factors and diseases must be shown in Table 1. What were the causes for AHF in the included participants, if not acute coronary syndromes (an exclusion criterion for this study, but among the most common causes worldwide): arrhythmias, myocarditis, pulmonary embolism, acute valvular dysfucntion, others...?

4. Tables style must be corrected to match the adopted journal style. 

Reviewer 3 Report

Comments and Suggestions for Authors

The authors examined the prognostic capacity of glomerular, tubular, and metabolic renal function in patients with acute heart failure. For this they used surrogate markers, eGFR, spot urine sodium, and serum bicarbonate, respectively. Based on the presence of the type of renal dysfunction the authors created 4-point renal dysfunction score and assessed the association with outcome. The data show a positive association between renal dysfunction score and 1-year mortality as well as a composite of 1-yaer mortality and HF rehospitalization.

The study is well-designed and conducted and etablished a valuble novel prognostic tool in AHF. 

However, there are some weaknesses that need to be addressed.

Major issues:

To provide more in depth analyses of the prognostic capacity of the defined renal prognostic score the authors should perform ROC curve analyses and examine whether the score improves prognostic capacity of NT-proBNP and cardiac troponin as well as of at least one established clinical score used for prognostication of mortality in AHF.  

The authors indicate in-hospital mortality as an endpoint but did not show any respective data.   

Did authors examine associations between covariates used for adjustment in multivariable Cox regression analyses and (Table 4) and 1-year mortality in univariable COX regression?

Why troponin I and not NT-proBNP was used as a covariate?

Sex should also be included as a covariate.

Could authors add data on NYHA class into Table 1 ?

Minor issues:

Please provide more methodologic details on NT-proBNP and cardiac troponin measurements.

Lane 129: A p-value; repetition, please delete.

Bicarbonate in the tables and text should be written as HCO3- .

Lanes 243-244: (Direct measurement….) This sentence is not fully correct; please improve it.

Lanes 312-326: Repetition; please delete.

Round 2

Reviewer 3 Report

Comments and Suggestions for Authors

The authors improved the manuscript.

The authors’ justification for not including the results of C-statistics, namely AUCs, is only partially to the point, as they claim in the conclusion that the established renal score aids patient stratification and prognosis. Since the calculated AUCs indicate moderate prognostic performance of the score the authors should modify the last sentence in their Abstract: ‘’The established renal score may aid patient stratification and prognosis’’.

Minors:

Table 3 should be re-arranged:

1. The existing text ‘’Profile of renal dysfunction (number of renal dysfunction)’’ should be above the columns: no renal dysfunction, mono-.., bi-…tri-..

2. The text: Prespecified endpoints should be above the first column on the very left of the Table.

Table 3 Suppl. Mat.:

Please replace the existing title (which is not correct) with: Associations between covariates used in multivariable model and prespecified endpoints.
